# In Vitro Pharmacological Activity, and Comparison GC-ToF-MS Profiling of Extracts from *Cissus cornifolia* (Baker) Planch

**DOI:** 10.3390/life13030728

**Published:** 2023-03-08

**Authors:** Nkoana I. Mongalo, Maropeng Vellry Raletsena, Rabelani Munyai

**Affiliations:** 1College of Agriculture and Environmental Sciences Laboratories, University of South Africa, Private Bag X06, Florida 0610, South Africa; 2Department of Agriculture and Animal Health, College of Agriculture and Environmental Sciences Horticulture Centre, University of South Africa, Private Bag X6, Florida 1710, South Africa

**Keywords:** *Cissus cornifolia*, antimicrobial activity, antiproliferative, anti-inflammatory, gas chromatography time-of-flight mass spectrometry (GC-ToF-MS)

## Abstract

*Cissus cornifolia* (Baker) Planch is traditionally used in South African traditional medicine (SATM) to treat a variety of human infections. The antimicrobial activity of extracts from *C. cornifolia* was investigated in vitro against a plethora of pathogenic microorganisms using the microdilution assay. The acetone extract exhibited a notable minimum inhibitory concentration (MIC) value of 0.10 mg/mL against *Mycoplasma hominis* and a further MIC of 0.20 mg/mL against *Candida parapsilosis*, *Streptococcus agalactiae*, *Pseudomonas aeruginosa*, and *Enterococcus faecalis.* In the antiproliferative assays, both the ethyl acetate and methanol extracts exhibited a potent inhibition of the MCF-7-21 cell line. In the anti-inflammatory assays, both the ethyl acetate and methanol extracts exhibited IC_50_ values of 15.59 and 15.78 µg/mL against Cyclooxygenase-2 (COX-2), respectively. Methanol extract further exhibited potent dual inhibition of both COX-2 and 15-LOX enzymes, hence, recommended to curb both related cancers, particularly breast cancer and inflammation-borne diseases. In the comparative gas chromatography time-of-flight mass spectrometry (GC/TOF-MS), the acetone, ethyl acetate, and methanol extract contained significantly prevalent amounts of compound 2-(2’,4’,4’,6’,6’,8’,8’-Heptamethyltetrasiloxan-2’-yloxy)-2,4,4,6,6,8,8,10,10-nonamethylcyclopentasiloxane with % area ranging from 15.714 to 39.225. The findings in the current work validates the use of the plant species in SATM in the treatment of cancer-like infections, opportunistic infections associated with HIV-AIDS. Furthermore, the in vivo studies and the mechanisms of action still need to be explored.

## 1. Introduction

South Africa has a rich biodiversity of plants which are mainly used as a first line of defense against various human and animal infections [1]. These medicinal plant species are becoming a center of attention to various researchers as pathogenic microbes are currently developing some resistance, especially in hospital setups, to a wide variety of antibiotics used mostly in developing countries to manage and treat infections [2]. This is further compounded by higher HIV-AIDS infections in Africa, which often result in the reactivation of the tuberculosis germ [3]. 

The genus *Cissus* comprises about 350 plant species distributed all over the world, mostly in tropical and subtropical regions. Some members of the genus are edible and serve as a source of both food and beverages while others are used as medicinal plants of great importance in treating a variety of both human and animal pathogenic infections [4,5,6]. Within Limpopo Province, South Africa, the bulbs are used in the treatment of various infections including cancer, sexually transmitted infections, and opportunistic infections associated with HIV-AIDS, and as a general well-being medicine [7]. Pharmacologically, some members of the genus are known to possess important activities which include antimicrobial, antioxidant, anti-diabetic, antimalarial, anti-inflammatory, protein binding capacity, hepatoprotective effect, anti-ulcer, anti-parasitic and other relevant properties [8,9,10,11,12]. *Cissus cornifolia* is a scandent shrub that belongs to the family Vitaceae and produces purplish fruits which are edible when ripe [13]. The plant species usually possess multiple stems, lateral roots which bear reddish bulbs, and leaves that are hairless on both sides, simple, alternate, and ovate to elliptic in shape [14]. The margins of the leaves are mostly toothed, while the flowers are yellowish and appear in clusters.

Cancer is an illness that results from the development of tumor cells surpassing the growth of normal human cells and may well infect important human body organs that include the breast, liver, stomach, cervix, brain, blood, and prostate [15]. The disease may be genetically inherited and is more prevalent in Europe than in any other continent. According to Bray et al., between 2008, and 2030, the disease is likely to increase to about 75% globally and is likely to double in less and underdeveloped countries worldwide [16]. Although surgery, radiation, chemotherapy, immunotherapy, hormone therapy, or gene therapy have been used as forms of treatment in Western methods of healing, there is a great need to discover plant-based anti-cancer medicine and possible compounds from such ethnopharmacologically tested plants with less or no side effects [17,18]. Inflammation is one of the most complex processes that are well-associated with pain, cancer, swelling, redness, heat, fever, and wound healing [19]. Besides helping in the maintenance of renal and gastric homeostasis, COX-1 is an enzyme that is localized and mostly prevalent in important essential organs of the human body, such as the kidney, stomach, and platelets, while COX-2 is an enzyme that presents in low amounts in the human kidney, brain, and ovaries [20]. The expression of COX-2 is largely induced in the case of mitogenic stimulation, inflammation, or tissue damage [21]. Lipoxygenase (15-LOX) is a lipoxygenase that could well react with polyunsaturated fatty acids and produces a variety of metabolites that are implicated in many important human diseases, such as cancer, and several respiratory diseases such as asthma, and chronic bronchitis [22].

The current work is aimed at investigating the in vitro antimicrobial activity of *Cissus cornifolia* bulb extracts against a plethora of important pathogenic microorganisms implicated as causative agents of sexually transmitted infections (*Candida albicans* and *Mycoplasma hominis*), urinary tract infections (*Escherichia coli*, *Pseudomonas aeruginosa*, *Proteus vulgaris*, *Enterococcus faecalis*, *Bacillus cereus*) and opportunistic infections associated with HIV-AIDS (*Cryptococcus neoformans* and *Candida parapsilosis*, *Staphylococcus aureus*, *Streptococcus agalactiae*), and those infecting the skin *(Moraxella catarrhalis*). Furthermore, the aim is to evaluate the anticancer activity against human hepatocellular carcinoma (HepG2), lung cancer (A547), cervical (HeLa), and breast cancer (MCF7-21), as the plant species is used in the treatment of cancer-related infections. HIV-AIDS and cancer are related to inflammation and pain; hence, the plant extracts were evaluated against *Soybean lipoxygenase* and both cyclooxygenase enzymes (COX-1 and 2) in vitro. The phytochemistry of the plant extracts was further compared using gas chromatography time-of-flight mass spectrometry (GC-ToF-MS). The potent in vitro biological activity of the plant species may well render the plant essential for use as a first line of defense against a plethora of infections, hence, serving as a complementary and alternative medicine to Western medicines which may not be accessible to communities in both underdeveloped and developing countries.

## 2. Materials and Methods

### 2.1. Plant Collection, Authentication, and Extraction

*Cissus cornifolia bulbs* were collected from Westphalia (ga-Broekman, GPS coordinates −23°08′21.12″ S 29°00′4.68″ E), adjacent to Phala secondary school, area within Blouberg Municipality (Limpopo Province, South Africa) in June 2021. The plant specimen was chopped using a sharp knife into small pieces, which were washed with running tap water to remove all the possible contaminants and adhering soil debris. The specimen was then packaged into newspapers and then transported to the Laboratory, UNISA Science campus, Eureka Building. The specimen was then placed on a laboratory bench to completely dry. Dried plant materials were ground into thin powder using a Scientec Hammer mill (Merck, South Africa). The voucher specimen (MNI-31) was also collected and then lodged at the University of South Africa (Science Campus, FL, USA) herbarium. 

The ground plant material was separately extracted with different solvents such as acetone, hexane, dichloromethane, ethyl acetate, and methanol (AR grade, Merck, South Africa) at a ratio of 1:5 *w*/*v*, and then kept in a shaker (Already Enterprise Inc., Taiwan, China, Model: LM-600 RD) for 72 hrs undisturbed. Samples were filtered using Whatman’s No. 1 filter paper and the resulting extracts were concentrated using *Buchi* Rotary evaporator (Bibby Scientific Limited, Stone, UK) at a reduced vacuum, connected to a Stuart vacuum pump (RE3022C) and Stuart recirculating cooler (SRC4) at 5 °C. The dried extracts were weighed, and the percentage yields were calculated accordingly. The extracts were then kept in a refrigerator at 4 °C until needed for all biological activities. All the experiments were carried out in accordance with the relevant guidelines and regulations (Ethical clearance: UNISA-Mongalo 90229436). 

### 2.2. Antimicrobial Activity

#### 2.2.1. Selected Microorganisms

Microorganisms such as *Moraxella catarrhalis* (clinical isolate from the sputum of HIV-AIDS patient), *Cryptococcus neoformans* (clinical isolate from wounds on the leg of HIV-AIDS patient), *Candida albicans* (isolated from the mouth of HIV-AIDS patient), and *Staphylococcus aureus* (isolated from the wound of HIV-AIDS patients) were selected for the study. Reference strains of *Streptococcus agalactiae* (12386), Enterococcus faecalis (19948), *Candida parapsilosis* (ATCC 22019), *Bacillus cereus* (10702), *Escherichia coli* (ATCC 25922), *Mycoplasma hominis* (ATCC 15488), *Pseudomonas aeruginosa* (ATCC 10031) and *Pseudomonas vulgaris* (ATCC 49132) were also selected for the study for comparison purposes. Bacterial strains were maintained on Muller Hinton agar slants, except *Mycoplasma hominis*, which was maintained on *Mycoplasma* agar supplemented with Mycoplasma G, while fungal strains were maintained on Potato Dextrose agar (PDA). All the growth mediums were obtained from Sigma-Aldrich (Schnelldorf, Germany).

#### 2.2.2. In Vitro Microdilution Assay

The antimicrobial activity of the extracts from *C. cornifolia* bulbs was investigated using the broth microdilution broth assay adopted from Eloff [23], with slight modification while the antifungal activity of the extracts was determined using the method previously described by Masoko et al. [24]. The overnight cultures of the selected microorganisms were standardized and diluted with freshly prepared broth to a concentration of 1.1 × 10^7^ cfu/mL. *Mycoplasma hominis* was grown in *Mycoplasma* broth (CM0403), supplemented with mycoplasma supplement G while all other bacterial strains were grown on Muller Hinton broth (MHB)(Oxoid). The fungal strains were grown on and maintained on Potato dextrose broth (PDB), from Merck Millipore, South Africa. 

In brief, a volume of 100 μL of the selected medicinal plant extracts at 25 mg/mL dissolved in 5% DMSO (Merck, South Africa), was added to a 96-well micro-plate containing about 100 μL of cooled sterile ultrapure distilled water, and then twofold serially diluted to yield different extracts concentrations. Bacterial culture of a standardized known concentration was aseptically transferred and added (100 μL) to each of the wells in a 96-well plate. Gentamicin was used as a positive control for all the selected bacterial strains while amphotericin B was used as the positive control for all the selected fungal strains. 5% DMSO was used as the negative control. Plates were then incubated at 37 °C overnight. Plates of all three fungal strains were incubated for 48 h, those of *Mycoplasma hominis* were incubated for 24 h and those of all other bacterial strains were incubated overnight. To each of the wells, 40 μL of 0.2 mg/mL freshly prepared *p*-iodo-nitrotetrazolium chloride (INT) dissolved in sterile distilled water was added in the 96-well plates and then incubated for 30 min at the same temperature, after which results were read. The MIC was defined as the lowest concentration of the extract that inhibits bacterial growth. The red color indicates the growth of a microorganism while the clear color indicates the concentration of the extract that inhibits the growth of the microorganism.

### 2.3. Anti-Inflammatory Activity

#### 2.3.1. Soybean Lipoxygenase (15-LOX) Assay

The anti-inflammatory activity of the extracts from *Cissus cornifolia* at a stock solution of 5 mg/mL was evaluated against the 15-LOX enzyme using the method explained previously by Pinto et al. [25]. The 15-LOX (Sigma-Aldrich, Germany) was made up of a working solution of 200 units/mL and kept on ice. A volume of 12.5 μL of each of the test samples or control (dissolved in pure DMSO) was added to 487.5 μL of 15-LOX in a 96-well micro-plate and incubated at room temperature for 5 min. After incubation, about 500 μL substrate solution (10 μL linoleic acid which was dissolved in 30 μL ethanol, made up to 120 mL with 2 M borate buffer at pH 9.0) was added to the solution. After another 5 min of incubation at room temperature, the absorbance was measured using a microplate reader at 234 nm (SpectraMax 190, Molecular Devices, Germany). Quercetin at a concentration of 1 mg/mL was used as a positive control, while pure DMSO was used as the negative control in the assay. The percentage enzyme inhibition of each extract compared with negative control as 100% enzyme activity was calculated using the equation below [26].
% Inhibition = OD_extract_ − OD_blank_/OD_negative_ control − OD_blank_ × 100%

The results were expressed as IC_50_ ± SE, where IC_50_ is the concentration of plant extract that inhibits 50% of the enzyme calculated from the graphs.

#### 2.3.2. Cyclooxygenase Assay

The inhibitory activities of each extract at 20 mg/mL in DMSO, serially diluted in a buffer yielding final concentrations of 200, 100, 50, and 25 µg/mL, were evaluated for anti-inflammatory activity against both COX-2 and COX-1 using a functional assay based on the formation of PGE_2_ which is a stable oxidation product resulting from COX oxidation of arachidonic acid, using a screening assay kit (Cayman Chemical, Item No. 701050) [27]. Shortly, 10 µL of the inhibitors at different concentrations were added to the different wells, followed by 150 µL of assay buffer, 10 µL of hemin, and 10 µL of the enzyme, either COX-1 or 2. The plate was incubated at 25 °C in a shaker at 80 rpm (Already Enterprise Inc., Taiwan, Model: LM-600 RD) for 5 min to allow mixing. Arachidonic acid (20 µL) was added to all the wells, thereafter, mixed by pipetting up and down thrice. The plate was then incubated again for 5 min at 25 °C, and the absorbance was read at 590 nm using a microplate reader (Varioshskan Flash, Thermo Scientific, Amsterdam, The Netherlands). DMSO was used instead of an inhibitor, as negative control, while celecoxib was used as a positive control. The percentages of inhibitions were calculated using the formula below [28].
% Cyclooxygenase Inhibition = [1 − A_t_/A_0_] × 100],
where A_t_ represents the absorbance of the test sample, while A_0_ represents the absorbance of the blank solution. Celecoxib (Sigma-Aldrich, Germany) was used as a positive control.

### 2.4. Anticancer Activity of Extracts against Selected Cell Lines

Four different cancerous cell lines such as human hepatocellular carcinoma (HepG2), lung cancer (A547), cervical (HeLa), and breast cancer (MCF7-21) cells were obtained from Sigma-Aldrich (Germany), with informed consent for use only for Research and Development, experimental trials only. The antiproliferative effects of various extracts from *Cissus cornifolia* bulbs were evaluated in vitro against selected cell lines in vitro using the tetrazolium-based colorimetric (MTT) assay [29]. The viable cell growth of each cell line after incubation of cells with the extracts dissolved in DMSO (Merck, South Africa) was determined. Cells of a sub-confluent culture of each of the cell lines were harvested and centrifuged (Eppendorf AG, Hamburg, Germany) at 2.0 rpm for 5 min, and then re-suspended in growth medium to yield 5 × 10^4^ cells/mL. The growth medium used was Minimal Essential Medium (MEM, Whitehead Scientific, South Africa) supplemented with 0.1% gentamicin (Virbac) and 5% fetal calf serum (Highveld Biological, South Africa). A cell suspension, of the above-mentioned concentration, of 200 µL was pipetted into each well of columns 2 to 11 of a sterile 96-well microtiter plate in a sterilized Laminar flow cabinet (Labotec, PTY LTD, South Africa). MEM (200 µL) was added to wells of columns 1 and 12 to minimize the “edge effect” and maintain constant humidity. The plates were then incubated for 24 h at 37 °C in a 5% CO_2_ incubator (Separation Scientific, South Africa), until the cells were in the constant exponential phase of growth. The MEM was aspirated from the cells, which were then washed with 150 μL phosphate-buffered saline (PBS, Whitehead Scientific) and replaced with 200 µL of the selected test plant extract at different concentrations ranging from 7.5 to 1000 µg/mL. The serial dilutions of the extracts were prepared in fresh MEM. The microtiter plates were incubated at 37 °C in a 5% CO_2_ incubator for 48 h undisturbed. Untreated cells and positive control (doxorubicin chloride, Pfizer Laboratories) were included in the assay. Each experiment was repeated three times independently. The LC_50_ (concentration of the plant extract that inhibited 50% of cell growth) values were determined from the obtained graphs of the concentration vs. % inhibition using the formula below.
Percentage cell inhibition = [1 − A_t_/A_0_] × 100]

Results were recorded as mean ± SE. 

### 2.5. GC-ToF-MS Analysis of Extracts from Cissus Cornifolia Bulbs

The separation of compounds from selected extracts with potent biological activity was performed using a method adopted by Mongalo et al. [28]. Briefly, the extracts were dissolved in acetonitrile (GC-MS grade, Sigma Aldrich, Germany) to the lowest concentration (1 mg/mL) and then the separation of compounds was performed on a gas chromatography (6890N, Agilent technologies network) coupled to Agilent technology inert XL EI/CI Mass Selective Detector (MSD) (5975B, Agilent Technologies Inc., Palo Alto, CA). The GC-MS system was coupled to a CTC Analytics PAL autosampler. Separation was performed on a non-polar DB-5MS (30 m, 0.25 mm ID, 0.25 µm film thickness capillary column. Helium was used as the carrier gas at a flow rate of 1 mL/min. The injector temperature was maintained at 250 °C. 1 µL of the sample was injected in spitless mode. The oven temperature was programmed as follows: 50 °C for 2 min, ramped up to 70 °C at a rate of 2 °C/min for 6 min, and finally ramped up to 320 °C at 20 °C/min and held for 5 min. The MSD was operated in a full scan mode and the source and quad temperatures were maintained at 230 °C and 150 °C, respectively. The transfer line temperature was maintained at 250 °C. The mass spectrometer was operated under electron impact mode at an ionization energy of 70 eV, scanning from 35 to 500 *m*/*z*. The compounds were identified using two different libraries (NIST 95 and WILLEY275) for compound matches.

## 3. Statistical Analysis 

Statistical analysis was performed using Graph pad Prism Version 7. In the GC-ToF-ms, the results were reported from two different libraries (NIST 95 and WILLEY275) for compound matches.

## 4. Results

The antimicrobial, anti-inflammatory and antiproliferative effects of extracts from *Cissus cornifolia* has been evaluated in vitro. Furthermore, GC-ToF-MS was carried out to compare the presence and abundance of the various compounds within the extracts with a noteworthy biological activity.

### 4.1. In Vitro Antimicrobial Activity

Extracts from *Cissus cornifolia* exhibited some varying degrees of bacterial inhibition in the microdilution assay (Table 1). The acetone extract exhibited a noteworthy antimicrobial activity yielding a minimum inhibitory concentration (MIC) value of 0.1 mg/mL while hexane extract exhibited a MIC value of ≥12.50 against seven different microorganisms that include all the selected gram-negative bacterial strains such as *Mycoplasma hominis*, *Pseudomonas aeruginosa*, *Moraxella catarrhalis*, *Escherichia coli* and *Proteus vulgaris*. Both the methanol and acetone extracts exhibited a MIC value of 0.20 mg/mL against *Candida parapsilosis* while ethyl acetate extracts yielded a MIC value of 0.39 mg/mL against *Cryptococcus neoformans*, *Candida albicans*, *Escherichia coli*, *Staphylococcus aureus*, and *Enterococcus faecalis*. Both hexane and dichloromethane extracts exhibited a weaker antimicrobial activity against three gram-negative bacterial strains such as *M. hominis*, *P. aeruginosa*, and *Proteus vulgaris* yielding MIC values of ≥12.5 mg/mL, while *Bacillus cereus* exhibited some similar resistance pattern to ethyl acetate, dichloromethane, and hexane extracts. Besides exhibiting a notable MIC value of 0.20 mg/mL against *Candida parapsilosis*, the methanol extract exhibited a weaker inhibition against *Proteus vulgaris* yielding a MIC value of ≥12.5 mg/mL. The extract further exhibited a MIC value of 0.39 mg/mL against *Cryptococcus neoformans*, *Escherichia coli*, *M. hominis*, *Staphylococcus aureus*, *Streptococcus agalactiae*, and *P. aeruginosa*. 

All the bacterial and fungal strains were susceptible to neomycin and amphotericin B, respectively. Acetone extract further exhibited a noteworthy antimicrobial activity against several bacterial strains, yielding a MIC value of 0.20 mg/mL against *E. faecalis*, S. *agalactiae*, and *P. aeruginosa*. The acetone extract exhibited the highest total activity (TA) value of 765.00 against *C. parapsilosis*, *P. aeruginosa* and *E. faecalis*, and *S. agalactiae*, while the methanol extract exhibited a TA value of 564.00 against *C. parapsilosis* (Figure 1). Furthermore, ethyl acetate extract exhibited TA value of 480.77 against 7 different pathogenic microorganisms such as *C. albicans*, *C. neoformans*, *E. coli*, *M. honinis*, *S. agalactiae*, and *S. aureus*.

### 4.2. Antiproliferative and Anti-Inflammatory Activity of Extracts from Cissus Cornifolia

The ethyl acetate extract exhibited the lowest IC_50_ value of 10.82 µg/mL against the MCF7-21 cell line while methanol extract yielded an IC_50_ value of 24.06 µg/mL against similar cell lines (Table 2). The acetone and dichloromethane extracts exhibited a weaker IC_50_ value of >1000 µg/mL against both HeLa and HepG2 cell lines, while the ethyl acetate extract yielded an IC_50_ value of 88.84 against the A547 cell line.

In the anti-inflammatory assays, dichloromethane extracts exhibited a noteworthy inhibition of both 15-LOX and Cyclooxygenase-1 (COX-1) yielding IC_50_ values of 44.12 and 8.08 µg/mL, respectively, while ethyl acetate and methanol extracts exhibited IC_50_ values of 15.59 and 15.78 µg/mL against Cyclooxygenase-2 (COX-2), respectively (Table 3). Hexane extract exhibited the lowest IC_50_ value of 5.87 µg/mL against COX-1.

Acetone, ethyl acetate, and methanol extract exhibited IC_50_ values of >200 µg/mL against COX-1, while hexane extract exhibited the lowest IC_50_ against similar enzymes.

### 4.3. Comparison GC-ToF-MS Analysis of Extracts from Cissus Cornifolia

The acetone, ethyl acetate, and methanol extract contained significantly prevalent amounts of one compound detected and identified as 2-(2’,4’,4’,6’,6’,8’,8’-Heptamethyltetrasiloxan-2’-yloxy)-2,4,4,6,6,8,8,10,10-nonamethylcyclopentasiloxane with % area ranging from 15.714 to 39.225 (Table 4). Furthermore, acetone and methanol extracts contained Spiro[2.4]hept-5-ene,5-trimethylsilylmethyl-1-trimethylsilyl- and Octanedioic acid, while ethyl acetate and methanol extract contained Butylated Hydroxytoluene (BHT) at 6.127 and 3.769% respectively. 

Acetone and ethyl acetate extracts yielded varying concentrations of similar compounds such as 3-Butoxy-1,1,1,7,7,7-hexamethyl-3,5,5-tris(trimethylsiloxy)tetrasiloxane, and two methyl esters such as 5,8,11-Heptadecatriynoic acid and 2,6-Nonadienoic acid.

## 5. Discussion

Antimicrobial resistance (AMR) of many pathogenic microorganisms renders the available antibiotics less efficient in treating many devastating human illnesses that may result in both morbidity and mortality [30,31]. According to Dadgostar [32], factors such as the overusing of different antimicrobial agents both in the healthcare setting as well as in the agricultural industry, the evolution, and mutation of bacteria, and passing the resistant genes through horizontal gene transfer are significant contributors to AMR. These factors are further compounded by the emergence of a plethora of opportunistic co-infections associated with HIV-AIDS, the emergence of several skin cancers, and tuberculosis [33,34]. Researchers have developed an enormous interest in addressing AMR challenges by using medicinal plants as a source of medicine, as it is well used in many rural African villages [35,36,37,38]. In a microdilution assay, the acetone extract exhibited a notable minimum inhibitory concentration (MIC) value of 0.10 mg/mL against *Mycoplasma hominis* and a further notable MIC of 0.20 mg/mL against *Candida parapsilosis*, *Streptococcus agalactiae*, *Pseudomonas aeruginosa* and *Enterococcus faecalis* (Table 1). These results may well suggest that the active phytocompounds from the plant species are highly soluble in acetone. These are corroborated by results from Musa et al. [39], who reported the acetone extract and an isolated compound, 4, 6-dihydroxy-5-methoxy-3-(1, 2, 3, 4, 5-pentahydroxypentyl)-2-benzofuran-1(3H)-one, to exhibit notable zones of inhibitions, ranging from 17 to 15 mm, against a plethora of microbes such as *Candida albicans*, *Bacillus subtilis*, *Staphylococcus aureus*, *Shigella dysenteriae*, *Streptococcus pyogens* and *Shigella flexineri*. These results are not comparable to the findings in the current study due to differences in methodologies used and selected bacterial strains. Although there is no standard benchmark on the potential antimicrobial extracts and characterized compounds, the consensus is that medicinal plant extracts yielding a MIC value of 0.1 mg/mL are notable and could be further evaluated for possible isolation of compounds and possible commercialization [40]. In the current work, the extracts with MIC values of 0.20 mg/mL or less are of potent antimicrobial activity, 0.21 to 0.40 mg/mL as moderate activity, and 0.41 mg/mL and above as inactive. These may well define acetone and methanol extracts as active against *Candida parapsilosis* which is known as an emerging major human pathogen prevalent in neonates and patients in intensive care units [41,42]. Other *Candida* species are prevalent opportunistic infections associated with HIV-AIDS, leading to various forms of candidiasis which are sexually transmissible and resistant to many antibiotics available in many developing and underdeveloped countries [43,44]. *Mycoplasma hominis* was the most susceptible microbe to acetone extract. A similar trend has been observed elsewhere [45]. Besides being implicated as a causative agent of septicemia, *Mycoplasma hominis* has long been known as a commensal of the women’s genital tract associated with various genitourinary tract infections and the complications of pregnancy [46,47]. It is important to note that the hexane extract exhibited poor antimicrobial activity (MIC ≥ 12.5 mg/mL) against gram-negative bacterial strains, hence, inactive. Contrarily, elsewhere, the hexane extracts from many bulbs exhibited potent antimicrobial activity against both gram-negative and gram-positive bacterial strains [48,49]. Although acetone extract exhibited a potent antimicrobial activity (MIC ≤ 0.20 mg/mL) against a plethora of microbes such as *C. parapsilosis*, *M. hominis*, *Pseudomonas aeruginosa*, *Enterococcus faecalis*, *Streptococcus agalactiae*, there is a need to explore the toxicological aspects and mode of antimicrobial activity against the selected microorganisms. It is important to note that microbes such as *S. agalactiae*, *P. aeruginosa*, and *E. faecalis* are implicated in ethnoveterinary infections including mastitis in dairy ruminants [50,51].

The maintenance of milking machines, hazard identification and critical control point (HACCP), and proper disinfection of other utensils used in milking are general measures to prevent new cases of mastitis [52,53]. Other authors have proposed the use of metabolomics as a tool to identify the biomarkers of mastitis as the key to controlling mastitis [54,55,56]. Worldwide, mastitis is well known to infect milk and result in major agricultural losses and various human infections resulting from ingesting infected milk. The current findings in the work support the use of *C. cornifolia* in the treatment and management of various human and animal infections, hence, they are relevant in South African traditional medicine (SATM). Besides potent antimicrobial activity, the total activity, depicted in Table 2, suggests that the acetone extract exhibited the highest TA ranging from 765.00 to 1530 mL/g against *Mycoplasma hominis Candida parapsilosis*, *Streptococcus agalactiae*, *Pseudomonas aeruginosa*, and *Enterococcus faecalis.* The total activity (TA) of a plant is the quantity of material extracted from one gram of dried and ground plant material divided by the minimum inhibitory concentration value [57]. TA indicates the largest volume to which the biologically active compounds in one gram of plant material can be diluted and still inhibit the growth of the test microorganism irrespective of the type and origin of a microbe [58]. This parameter is important when evaluating the potential use of plant extracts for treating various devastating and health-threatening microbial infections [59]. The acetone extracts of *C. cornifolia* exhibited the highest total activity compared to the other extracts (Figure 1). Thus, one gram of *C. cornifolia* acetone bulb extract can be diluted to 1.5 L with water and still inhibit the growth of *M. hominis*, while a similar acetone extract could be diluted with 0.76 L of water and still inhibit the growth of *Candida parapsilosis*, *Streptococcus agalactiae*, *Pseudomonas aeruginosa*, and *Enterococcus faecalis.* The higher the TA, the higher the safety margin [60]. However, the cytotoxicity of the plant extracts against a plethora of human cell lines needs to be explored, both in vitro and in vivo.

Although surgery, radiation, chemotherapy, immunotherapy, hormone therapy, or gene therapy have been used as forms of treatment in Western methods of healing, there is a great need to discover plant-based anti-cancer medicine and possible compounds from such ethnopharmacologically tested plants with less or no side effects [61]. In the assay antiproliferative assay (Table 2), the decrease in cell viability indicates a diminished capacity in mitochondrial dehydrogenase to convert the yellow tetrazolium salt into some purple crystal formazan [26].

Methanol and ethyl acetate extracts selectively inhibited the growth of the MCF-7 cell line, compared to other cell lines, yielding significant IC_50_ values of 24.06 of 10.82 µg/mL, respectively. The lower IC_50_ after treatment with ethyl acetate and methanol extracts indicates the sensitivity of breast cancer cell lines when exposed to treatment with such extracts. According to the United States Cancer Institute, a plant extract is referred to as inhibitory to cancer cell lines if exhibiting an LC_50_ of less than 20 μg/mL [62]. These may well suggest that only the ethyl acetate extract satisfies that criterion, hence, worth introspecting for further analysis. In the current work, the extracts with LC_50_ values ranging from >20 μg/mL to 45 μg/mL are moderately exhibiting some cytotoxic effect or anticancer activity against the tested cell line of choice, while extracts exhibiting IC_50_ > 45 are inactive [15]. According to Mongalo and Makhafola [7], some traditional healers within the Blouberg area, Limpopo Province, South Africa, topically apply the powdered plant material directly to the wound or the infected breast to treat and manage breast cancer. The plant species are used in combination with other species such as *Bulbine anguistifolia* Poelln. (Marumo a ngata) and *Sarcophyte sanguinea* subsp. *piriei* (Hutch.) B. Hansen. Otherwise, the plant material is cooked and drunk and taken orally three times to alleviate cancer and breast-related infections. Although these plant species are recommended for such treatments, there are no reports from the literature on the toxicological aspects of the plant species. There is also a need to perform cytotoxicity studies of the extracts from the plant species against normal human cell lines both in vitro and in vivo. It is also important to note that in vitro studies do not always equate to in vivo studies [63]. According to Saeidnia et al. [64], the use of animal models provides some drawbacks such as the difference in biokinetics parameters or extrapolation of results to humans, hence, they are more reliable than in vitro tests. Various forms of cancer are well-associated with some inflammatory infections and may well involve pain, swelling, and other forms of inflammation. Inflammation is a generally recognized hallmark of cancer that substantially contributes to the development and progression of a disease (cancer), resulting from uncontrolled growth and persistent division of the cancerous cell lines resulting in tumors [65].

The extracts from *C. cornifolia* were evaluated for anti-inflammatory activity in vitro against Soybean lipoxygenase (15-LOX) and two cyclooxygenase enzymes, such as COX-1 and COX-2 (Table 3). Although the hexane and dichloromethane extract exhibited a noteworthy and potential IC_50_ value of 5.87 and 8.08 µg/mL against COX-1, it is not of paramount importance as the extracts inhibiting COX-1 are likely to tamper negatively with the functioning of human essential organs and further result in hemorrhage, kidney failure, and or severe side effects on the gastrointestinal area [66]. In accordance with the 15-LOX and COX-2 assays, our results also displayed that only the methanol extract significantly exhibited dual inhibition of both 15-LOX and COX-2 yielding IC_50_ values of 68.88 and 15.78 µg/mL, respectively. A plant-based extract is believed to have a potent inhibitory effect only if it exhibits an IC_50_ value of ≤100 µg/mL, against both COX-2 and 15-LOX enzymes [67]. According to Meshram et al. [60], such dual inhibitors could well inhibit the release of both prostaglandins and leukotrienes from the respective cyclooxygenase and lipoxygenase pathways, hence, they are recommended for use as both anti-inflammatory and anticancer agents. Considering the role of 15-LOX and COX-2 in the progression of some cancers, the discovery of medicinal plant species as dual inhibitors could potentially lead to the development of novel cancer therapeutics, and it can be claimed that 15-LOX inhibitors might also be suitable candidates as possible chemotherapy agents soon [68]. Such inhibitors could well assist in curbing neurological diseases such as Parkinson’s and Alzheimer’s diseases, amongst other devastating human infections [69]. Furthermore, although methanol extract exhibited the most important dual inhibition of both 15-LOX and COX-2, the specific metabolites that are responsible for this activity and the possible mechanism of action remain unknown and need to be further explored. There is also a need to explore the in vitro anti-inflammatory activity of the plant extracts against pro-inflammatory enzymes such as IL-1, IL-6, TnF-ά, and others. Furthermore, the in vivo activities need to be explored.

In comparing the phytocompounds from three extracts, viz acetone, methanol, and ethyl acetate extracts, which exhibited potent biological activity, only an organoheterosilane compound known as 2-(2’,4’,4’,6’,6’,8’,8’-Heptamethyltetrasiloxan-2’-yloxy)-2,4,4,6,6,8,8,10,10-decamethylcyclopentasiloxane was detected in all extracts (Table 4). A similar compound was earlier identified from another medicinal plant extract with potent antibacterial activity [15,70]. Although the biological activity of the compound is not well documented, the organoheterosilanes are well known for their potential antimicrobial and anti-inflammatory activity [71,72]. However, the mode of action and the mechanism of action thereof are not well documented. However, the organoheterosilane compounds are likely to inhibit bacterial growth by preventing the formation of links between the peptidoglycan layers making the bacterial structure unstable, thereby allowing antibiotic entry and sudden death of cells [73]. Acetone and ethyl acetate extracts yielded varying concentrations of similar compounds such as 3-Butoxy-1,1,1,7,7,7-hexamethyl-3,5,5-tris(trimethylsiloxy)tetrasiloxane, and two methyl esters such as 5,8,11-Heptadecatriynoic acid and 2,6-Nonadienoic acid, while acetone and methanol extract contained Spiro[2.4]hept-5-ene,5-trimethylsilyl methyl-1-trimethylsilyl- and Octanedioic acid, while ethyl acetate and methanol extract contained Butylated Hydroxytoluene (BHT) at 6.127 and 3.769%, respectively. Methyl esters are well known for their selective inhibition of MCF-7 cancerous cell lines in vitro [74]. Such methyl esters may directly downregulate the Bcl-2 protein expression in cells [75]. Spiro[2.4]hept-5-ene,5-trimethylsilyl methyl-1-trimethylsilyl-, and Octanedioic acid have been reported to possess enormous antimicrobial activity against several human and animal pathogens in vitro [76,77,78,79]. Ethyl acetate and methanol extract exhibited a reasonable quantity of (BHT) Butylated Hydroxytoluene (Table 4). BHT was detected in plenty of fruits, vegetables, and a variety of medicinal plants worldwide and has been reported to possess a wide variety of pharmacological activities, particularly antioxidant activity [80,81]. These corroborate our results as only methanol and ethyl acetate exhibited 3.769 and 6.127% of BHT, respectively, and a potent anti-inflammatory activity yielding potent IC_50_ values of 15.77 and 15.59 µg/mL, respectively. The current work validates the use of the plant species in SATM. According to Nguyen et al. [66], plant samples containing relatively higher quantities of BHT are likely to exhibit anti-inflammatory activity through ROS inhibition. Furthermore, the mode of action of BHT in some in vivo studies is related to dose-diminution of UV radiation dose reaching respective target sites [62]. In the antimicrobial activity, like in many other antibiotics, the compounds identified may well result in targeting protein synthesis, DNA replication, the cell membrane, and the cell wall in the bacteria [72].

## 6. Conclusions

The extracts from *Cissus cornifolia bulbs* had varying degrees of antimicrobial, anti-inflammatory, and anticancer activities in vitro. The mechanism of action of such extracts against the selected microorganisms remains unknown. To the best of our knowledge, this is the first work to report the comprehensive phytoconstituents from *C. cornifolia*. Although the biological activity observed may well validate the use of the plant species in the treatment and management of a variety of infections, there is a need to explore the in vivo studies. This is prompted by the biological activity observed. The biological activity observed may well be attributed to both the less and most abundantly detected compounds individually or synergistically. For that reason, there is also a need to explore the biological activity of these compounds in combinations.

## Figures and Tables

**Figure 1 life-13-00728-f001:**
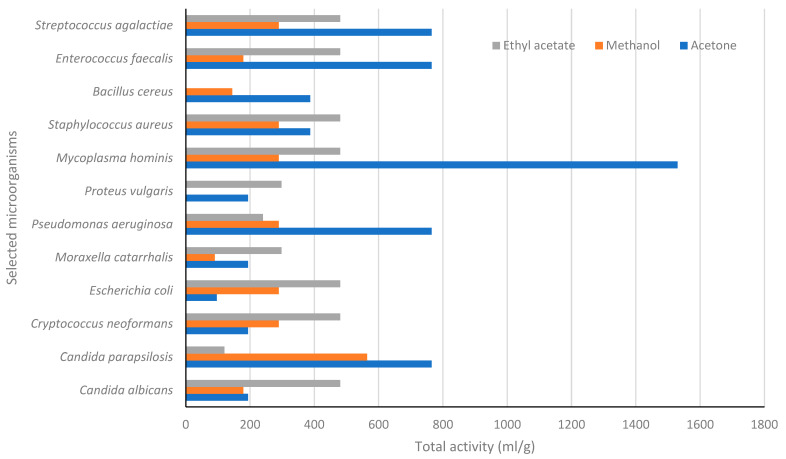
The total activity (mL/g) of various *Cissus cornifolia* extracts against selected microorganisms.

**Table 1 life-13-00728-t001:** Antimicrobial activity of extracts (MIC in mg/mL) from *Cissus cornifolia* bulbs.

Microorganisms	Extracts and Control Drugs					
	Acetone	Methanol	Dichloromethane	Ethyl Acetate	Hexane	Control Drugs
Yeasts						Amphotericin B
*Candida albicans*	0.78	0.63	0.16	0.39	0.31	0.001
*Candida parapsilosis*	**0.20**	**0.20**	≥12.50	1.56	1.56	0.004
*Cryptococcus neoformans*	0.78	0.39	1.25	0.39	0.39	0.001
Gram-negative bacteria						Neomycin
*Escherichia coli*	1.56	0.39	0.39	0.39	≥12.50	0.001
*Moraxella catarrhalis*	0.78	1.25	1.25	0.63	≥12.50	0.013
*Proteus vulgaris*	0.78	≥12.50	≥12.50	0.63	≥12.50	0.013
*Pseudomonas aeruginosa*	**0.20**	0.39	≥12.50	0.78	≥12.50	0.008
*Mycoplasma hominis*	**0.10**	0.39	≥12.50	0.39	≥12.50	0.008
Gram-positive bacteria						
*Staphylococcus aureus*	0.39	0.39	0.39	0.39	≥12.50	0.003
*Bacillus cereus*	0.39	0.78	≥12.50	≥12.50	≥12.50	0.004
*Enterococcus faecalis*	**0.20**	0.63	0.63	0.31	0.625	0.008
*Streptococcus agalactiae*	**0.20**	0.39	≥12.50	0.39	1.56	0.004

Bold-faceted data shows a noteworthy antimicrobial activity.

**Table 2 life-13-00728-t002:** Antiproliferative effect (IC_50_ in µg/mL) of extracts from *C. cornifolia*. The letters represent the statistical significance, when comparing the means of the treatments.

Extracts	HeLa	MCF7-21	HepG2	A547
Hexane	119.67 ± 0.09 ^b^	314.81 ± 0.13 ^c^	138.43 ± 0.18 ^d^	104.33 ± 1.11 ^c^
Acetone	>1000	161.85 ± 0.09 ^b^	>1000	421.22 ± 1.89 ^d^
Ethyl acetate	123.83 ± 0.98 ^bc^	10.82 ± 0.04 ^a^	113.21 ± 0.98 ^bc^	88.84 ± 0.11 ^c^
Methanol	103.89 ± 0.11 ^c^	24.06 ± 0.01 ^a^	96.45 ± 0.16 ^d^	>1000
Dichloromethane	>1000	162.55 ± 0.12 ^c^	>1000	>1000
Doxorubicin	1.43 ± 0.04 ^a^	1.26 ± 0.03 ^a^	1.11 ± 0.02 ^a^	2.10 ± 0.01 ^a^

**Table 3 life-13-00728-t003:** Anti-inflammatory activity (IC_50_ in µg/mL) of extracts from *C. cornifolia*. The letters represent the statistical significance, when comparing the means of the treatments. That is a, very closely related and e, far apart.

Extracts	15-LOX	COX-1	COX-2
Hexane	188.13 ± 1.22 ^e^	5.87 ± 0.09 ^b^	>200
Acetone	125.88 ± 0.01 ^a^	>200	120.22 ± 0.01 ^a^
Dichloromethane	**44.12 ± 0.16 ^d^**	8.08 ± 0.04 ^a^	58.77 ± 0.04 ^a^
Ethyl acetate	164.36 ± 0.15 ^d^	>200	**15.59 ± 0.02 ^b^**
Methanol	**68.88 ± 0.11 ^bc^**	>200	**15.78 ± 0.01 ^a^**
Quercetin (µg/mL)	31.55 ± 0.01 ^a^	-	-
Celecoxib (µM)	-	12.88 ± 0.01 ^a^	4.33 ± 0.01 ^a^

Bold faceted data shows a noteworthy anti-inflammatory effect.

**Table 4 life-13-00728-t004:** GC-ToF-MS analysis of acetone, ethyl acetate, and methanol extracts from C. cornifolia.

Extracts	RT min:s	Compound Detected	Similarity	Area %
Acetone	09:86,1	5,8,11-Heptadecatriynoic acid	664	9.454
	11:39,5	2-(2’,4’,4’,6’,6’,8’,8’-Heptamethyltetrasiloxan-2’-yloxy)-2,4,4,6,6,8,8,10,10-nonamethylcyclopentasiloxane	936	15.714
	13:09,0	3-Butoxy-1,1,1,7,7,7-hexamethyl-3,5,5-tris(trimethylsiloxy)tetrasiloxane	756	12.281
	15:33,50	2-Pyrrolidinone, 1-methyl-	955	6.344
	16:16,0	Hexadecane	961	4.496
	17:17,9	Octanedioic acid	594	2.207
	17:36,4	2,6-Nonadienoic acid	627	2.227
	19:15,9	Spiro[2.4]hept-5-ene,5-trimethylsilylmethyl-1-trimethylsilyl-	779	1.806
Ethyl acetate	09:50,5	9-Borabicyclo[3.3.2]decan-10-ol, 9-(1-oxopropoxy)-, propanoate	591	6.127
	12:26,7	2-(2’,4’,4’,6’,6’,8’,8’-Heptamethyltetrasiloxan-2’-yloxy)-2,4,4,6,6,8,8,10,10-decamethylcyclopentasiloxane	935	39.225
	12:44,8	3-Butoxy-1,1,1,7,7,7-hexamethyl-3,5,5-tris(trimethylsiloxy)tetrasiloxane	757	10.275
	15:01,6	Butylated Hydroxytoluene	932	6.127
	16:38,1	5,8,11-Heptadecatriynoic acid	674	5.999
	16:55,5	2,6-Nonadienoic acid	620	2.084
	17:22,1	Quinoline	778	1.88
Methanol	09:39,3	Spiro[2.4]hept-5-ene, 5-trimethylsilylmethyl-1-trimethylsilyl-	701	1.36
	11:50,5	2-(2’,4’,4’,6’,6’,8’,8’-Heptamethyltetrasiloxan-2’-yloxy)-2,4,4,6,6,8,8,10,10-nonamethylcyclopentasiloxane	806	19.478
	12:44,7	Trisiloxane, 1,1,1,5,5,5-hexamethyl-3,3-bis[(trimethylsilyl)oxy]-	887	17.705
	15:18,5	Butylated Hydroxytoluene	934	3.769
	15:39,0	1,6-Heptadiene	654	4.442
	16:55,9	1,3-Dioxolane	883	2.224
	17:13,2	Octanedioic acid	551	1.064

## Data Availability

The data supporting the article is kept with authors and available on request.

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
