# Peer review of "In Vitro Pharmacological Activity, and Comparison GC-ToF-MS Profiling of Extracts from Cissus cornifolia (Baker) Planch"

_life, 2023, doi:10.3390/life13030728_

Round 1

Reviewer 1 Report

·      I suggest improving the title, it has a lot of content, beneficial activity and compound profile.

·      In the abstract it has a lot of results, it significantly improves the conclusion and the last sentence doesn't make sense to go.

·      I suggest changing that phrase “is blessed with” in line 37.

·      I suggest significantly improving the end of the introduction, from line 58-72, there is a lack of order, the objective lacks clarity.

·      I suggest checking if it is necessary to place what you have in the methodology, from line 83-89. I think it is better to quote that information and not place it.

·      The sentence of line 209-212 seems to be a result that is in materials and method, I suggest checking.

·      Check that the tables are cited correctly in the text, at least 1 does not appear where it should be.

·      In paragraph 243-260 it repeats a lot of information that is in table 1, I suggest improving that part. this is repeated in various parts of the results.

·      Phrase of line 264-266 is methodology.

·      Table 2 and 3 must not be followed.

·      The tables are missing the legends, they do not have enough information. Now these tables are not self explanatory.

·      I suggest not placing table 5, 6 and 7 continuous, better to make a single table for that information.

·      I suggest changing some of those tables from the methodology to graphics, to improve the manuscript.

·      I suggest removing the phrase from line 331-332, if it is not comparable with the work, it is not necessary to place it.

·      In the discussion I consider that it exceeds some sentences that should go in the introduction, for example the phrase 382-390 is too long. From line 421 forward.

·      In these extracts there are several metabolites in the medium from plant material and the beneficial effects cannot be attributed only to those that were identified or to others that are not part of the study.

·      There is a lack of clarity in the discussion.

·      The discussion is poor.

·      In general terms, the discussion could be better. I suggest adding more contributions from the researcher with the findings. The importance or significance of these results for science.

·      I suggest improving the connection between the different parts of the manuscript.

     It is necessary to improve the presentation of the work, order ideas, improve clarity and you must conclude your research.

·      I DID NOT FIND THE CONCLUSION OF THE MANUSCRIPT.

Author Response

The authors are thankful for the comment and have addressed all the sections where the Reviewer required amendments. Thank you for your inputs.

We truly appreciate it.

Reviewer 2 Report

I recommend the article entitled ,, Antiproliferative, antimicrobial, anti-inflammatory activity, and comparison GC-ToF-MS profiling of extracts from Cissus cornifolia (Baker) Planch from South Africa” for publication in the Life, but after some corrections. The article is well written, but in my opinion a clearly formulated research objective should be added and the innovative nature of the research should be highlighted.

Some minor mistakes:

Line 51: correct the parenthesis from ( to [

Line 183: bold should be removed

Some major mistakes:

Why you used student’s t-test for statistical analysis? This test should only be used when two groups are compared.

Author Response

The authors are thankful for the comments and have addressed all the comments from the Reviewer. we truly appreciate your efforts. Thank you once more.

Reviewer 3 Report

1.In page 1, line 11-32

There are some problems with the part "Abstract"

First: the authors give very short and brief data about Cissus cornifolia (Baker) Planch which is not enough to introduce the importance of this item to other readers

Second: there is alot of detailed information about the results of the manuscript in this part. Thus, it can reduce the tendency of other reader to read other parts of the manuscript.

Third: there is a lot of results in the form of various numbers in this part. This is not suitable for the part" Abstract" to have alot of numbers about results. Authors should mentioned the total form of findings and maintain detailed data in the main text so that the curiosity of other readers to read other parts of manuscript can be kept

Forth: why authors have not write some brief sentences about their conclusion in this part?

2.In page 1, line 37-38

Why the sentence "South Africa is ... and animal infections" has not proper reference?

3. In page 1 and 2, line 43-46

Why authors have given detailed information about the structure of Cissus cornifolia? 

4.in page 1 and 2, line 43-46 and line 48-58

The authors have talked about Cissus cornifolia and after that, they have mentioned some information about The genus Cissus.

This has interrupted the consistency of the text. Please correcr this problem.

5.In page 2, line 58

Please correct the multiple reference in this part ([8-17])

6. In page 2, line 58- 72

Please reconsider this part according to notes below:

First: line 58-61, sentence"The current work is ... various in vitro assays" : this sentence belongs to the end of the part " Introduction"

Second: line 61-65, sentence" Plant extracts with potent ... spectrometry (GC-ToF-MS). "

Why authors have mentioned this sentence here?

Third: line 67-72

Authors have spoken about some biological effects of South African medicinal plants i this part. But this paragraph has disrupted the continuity of the text of "Abstract"

Please omit this paragraph or transfer it to its auitable place

Forth: line 68 and line 72 have multiple references. Please reform them.

7.Page 1 and 2, the part "Introduction" has some major lackages including:

First: lack of continuity and consistency

Second: lack of enough scientific data about different topics that authors have been trying to talk about

Third: authors could not mention the importance of their present work in this part

Forth: authors have not remark the purpose of their study clearly at the end of this part

Thus, the part "Introduction" should be reconsidered completely based on notes above

8. In page 2, line 76-78, part"2. Materials and Methods"

Please write geographical coordinates of the location that you have collected your specimens.

9. In page 2, line 80-81

The authors have mentioned that the specimen dried immediately, to avoid 

decaying, on the laboratory bench for three weeks.

This sentence seems to be a little confusing because the authors have mantioned that the process of drying of specimen was done "immediately" but "for three weeks".

Please wite this sentence more obvious.

10. In page 3, line 98-99, the authors have said " All the experiments were carried out in accordance with the relevant guidelines and regulations" but they have not mentioned any scientific reference for any step of this part of manuscript (part 2.1 Plant collection, authentication, and extraction)

11. Why authors have not provide an MTT-assay result and a proper graph based on the analysis of the results of MTT-assay?

12. In page 9, line 312-321

Why authors have written about antimicrobial 

resistance (AMR) in the part "Discussion"? Was it necessary? If yes, why?

13. In page 9, line 321-322

Why authors have remarked that" In the current work, the in vitro antimicrobial activity of various extracts from Cissus cornifolia bulbs was investigated against a plethora of microorganisms"

 This sentence should not be in this part. Please omit it.

14. In page 10, line 338 and 342

Please reconsider multiple references.

15. In page 10, line 355-363

Why authors have written about mastitis? What is the necessity of the explaination about mastitis in this part of the manuscript?

16. In page 10, line 370-372 

Appropriate references should be added at 

the end of each sentence

17. In page 11, line 382-390

This part belongs to the part" Introduction"

Please remove this section from the part" Discussion"

18. In page 11, line 390-393

The authors have mentioned this section in the "material and methods". Thus, this section should not be in the part "Discussion"

Please reconsider it.

19. Please add necessary reference at the end of the sentences in page 11, line 316-318

20. In page 11 line 416-420

This part is about inflammation. Please add it to the next paragraph and combine it properly with mentioned part.

 21. In page 11, line 427-428

Please consider suitable rederence for this part.

22. Why authors have not mentioned biological activities of the compounds found in GC-ToF-MS analysis of the extract from C. cornifolia in Tables 5, 6 and 7? Specially their anti-cancer, anti-inflammatory and anti-microbial activities

23. In page 11, line 453-481

Why authors have not talked about the molecular mechanisms of each molecule that is found in the extract from C. cornifolia?

(Specially the mechanism of anti-cancer, anti-inflammatory and anti-microbial acticities.

24. There is no "Conclusion" part in the manuscript. The authors should remark the reason of the absence of the part "Conclusion" in their manuscript.

25.Please check references carefully (especially their titles and dois)

26. Please exert the notes below about citation all over your manuscript

a) each sentence needs proper reference, thus, please reform all multiple references all over your manuscript 

b) add proper reference at the end of each sentence (exept sentences that are well-established or the results of present panuscript)

c) some citaions locate in the middle of some sentences Please ransfer these kind of citaions to the end of their related sentence.

Author Response

Good day Prof.

The Reviewers comments were well received and adressed.

Tha authors have attached the corrected version of the manuscript and believe it is well adressed. Thank you for your valued comments.

Round 2

Reviewer 3 Report

I could not find any responces to my comments and also corrections are not highlighted in the text.